# Study Regarding the Monitoring of Nocturnal Bruxism in Children and Adolescents Using Bruxoff Device

**DOI:** 10.3390/diagnostics13203233

**Published:** 2023-10-17

**Authors:** Adriana Elena Crăciun, Diana Cerghizan, Kinga Mária Jánosi, Sorin Popșor, Cristina Ioana Bica

**Affiliations:** Faculty of Dentistry, George Emil Palade University of Medicine, Pharmacy, Science, and Technology of Targu Mures, 38 Gh. Marinescu Str., 540142 Targu Mures, Romania; adriana.craciun@umfst.ro (A.E.C.); kinga.janosi@umfst.ro (K.M.J.); sorin.popsor@umfst.ro (S.P.); cristina.bica@umfst.ro (C.I.B.)

**Keywords:** bruxism, children, parafunctions, Bruxoff

## Abstract

Bruxism is a parafunctional activity represented by the gnashing and clenching of one’s teeth. The aim of this study was to determine the utility of screening and monitoring with a Bruxoff device during nocturnal bruxism in 51 children and adolescents (36 with bruxism and 15 without bruxism) by assessing the variations in the intensity and duration of parafunctional activity in each patient. Bruxoff measurements were recorded for at least 60 min for three consecutive nights for each subject. All the parameters recorded using Bruxoff in the control and the study groups showed a statistically significant difference (*p* < 0.05). The differences found by comparing the values recorded in the male and female study groups are significant for heart rate, the number of masseter muscle contractions during one night, and mixed contractions. The Bruxoff device proved to be important in diagnosing patients with bruxism in our practice.

## 1. Introduction

Bruxism is a parafunctional activity represented by the gnashing and clenching of one’s teeth. Another definition of nocturnal bruxism was established in 2018 within the International Consensus on the Assessment of Bruxism, according to which sleep bruxism (SB) is a masticatory muscle activity during sleep that is characterized as rhythmic (phasic) or non-rhythmic (tonic) and is not a movement disorder or a sleep disorder in otherwise healthy individuals [1]. This habit destroys teeth and leads to dental abrasion over time. Bruxism can be nocturnal or diurnal depending on the circadian moment when this parafunction occurs [2].

Nocturnal bruxism is a parafunctional disorder with multifactorial etiology, which occurs during the night [3]. Early bruxism diagnosis is essential because it can severely affect the quality of life [4]. A clinical diagnosis of nocturnal bruxism is based on the diagnostic criteria proposed by the American Academy of Sleep Medicine (AASM) [5].

Epidemiological studies have shown bruxism emerges in all age groups, but more often in young people [2]. The literature reports prevalence rates of bruxism in children ranging from 14% to 20%, while in adults, it varies between 6% and 8%, and decreases with age [6,7].

The diagnosis of bruxism is a significant challenge for dentists. Early diagnosis provides a perspective on control, prevents damage to the components of the masticatory system, and provides comfort [8,9].

Night-time polysomnography (PSG) with audio/video recording remains the gold standard for diagnosing nocturnal bruxism [10]. However, several electromyographic (EMG) recording devices have been introduced to detect bruxism episodes at home [11,12]. They have the advantage of being less expensive and easier to use than the polysomnographic method. In addition, screening at home helps to collect more representative data than recording in a sleep laboratory. This method is easy and it enables multiple-night recordings.

Bruxoff (Bruxoff^®^, Spes Medica, Battipaglia, Italy) is a four-channel portable device; three channels use surface EMG to monitor both masseter muscles and one channel to monitor heart rate. Previous studies have shown the validity of Bruxoff as a portable screening device for subjects with symptoms specific to nocturnal bruxism [11].

Due to the multifactorial etiology of bruxism, many treatment methods have been proposed, such as drug treatments, psychological treatment, and dental treatments. Dental treatments are divided into two categories: reversible treatments (the use of dental braces or occlusal stabilizers) and irreversible (orthodontic treatments, selective occlusal grinding, and surface dental restorations). Although there are several directions of treatment, none have been proven to be more effective than the others [9].

Our study aimed to determine the utility and validity of screening and monitoring with a Bruxoff device during nocturnal bruxism in young patients by assessing the variations in the intensity and duration of parafunctional activity in each patient with a view to establish a diagnosis from an early stage and to apply an early treatment.

## 2. Materials and Methods

This prospective study was conducted over 6 months, from 4 March to 4 September 2022. The study was in line with the provisions of the Declaration of Helsinki and approved by the Institutional Ethics Committee of the George Emil Palade University of Medicine, Pharmacy, Science and Technology of Targu Mures (1639/3 March 2022) for studies involving humans. The participants were informed in advance about our research’s purpose and had the possibility to withdraw from the study at any time. We detailed the study aims to all the participants and their parents or legal representatives. The study did not represent a risk and did not have side effects on the health of the participants. Informed consent was obtained from all subjects regarding their participation in this research, and each patient’s written informed consent statement was obtained to publish this paper.

The participants were selected based on a self-designed questionnaire that contained 9 questions, 7 closed questions and 2 open questions concerning specific signs and symptoms of nocturnal bruxism, the presence of vicious habits, as well as life events that could have an emotional impact on the child’s life and generate stress. This questionnaire was designed to build the study group represented by patients with bruxism and the control group represented by patients who did not present this parafunction, from the total group of children according to the specific inclusion and exclusion criteria. The children, or the parents/legal guardians of participants under the age of 18, answered the questions in the questionnaire.

The participants included in this study were aged 5–18 years old, belonging to both genders, and presenting specific signs and symptoms of bruxism according to the questionnaire. Patients with damage to the central nervous system, pacemaker wearers, those medicated with benzodiazepines, neuroleptics, or l-dopamine, and patients with known general disorders with implications for the functionality of the masticatory neuromusculature were excluded. 

The sample size was calculated based on the standard deviation (SD) and mean (M) from a similar study by Vladuțu et al. [13] using a web-based sample size calculator. Their study group consisted of 20 participants. Parameters: SD was 37.850, M was 23.374, the alpha level was set at 0.05, and the power of the test was set at 0.8.

A total of 51 participants (36 with nocturnal bruxism and 15 for the control group) were selected based on the bruxism prevalence evaluation questionnaire and the signed informed consent of the patients’ legal guardians. The selected patients were monitored via recordings using the Bruxoff^®^ portable device (Spes Medica in Battipaglia, Italy). This device consists of three channels. Two channels were used to obtain bilateral sEMG (surface electromyography) data from the masseter muscle, and a third channel was used for the heart rate (HF). The recordings were stored on a MicroSD card with a binary file embedded into the device. This device uses concentric electrodes (Code^®^, Spes Medica, Battipaglia, Italy), which could represent a methodological solution to improve the quality of the sEMG signals detected from masseter muscles using portable devices. The geometry of these electrodes allows for easy application by patients as they are invariant to rotations and reduce crosstalk phenomena due to their Laplacian design [14]. 

Single-use bipolar concentric electrodes called Code^®^ (Spes Medica in Battipaglia, Italy) [14] were used to detect the surface EMG signals from the masseter muscle on both sides. These electrodes have a radius of 16 mm, with the detection site at the cheeks (Figure 1a). The size of the inductive coil is important for the practicability of this system.

The heart rate was detected using a single-use bipolar electrode on the left side of the chest, just below the pectoral muscles (Figure 1b).

The EMG and HF signals were recorded overnight (during at least 3 h of sleep per night). The participants used the device and the installed electrodes at home, with or without technical assistance. All the patients received written and verbal instructions regarding the recording procedure and a telephone number to call in case of difficulties during the recordings (Figure 1c).

Each patient wore the Bruxoff monitoring device for three nights in a row for a minimum of 60 min.

According to the user manual for the device, the participants were instructed to perform three maximum voluntary contractions (MVCs) at the beginning of the recording for three seconds each and separated by approximately 10 s of rest [14]. The recordings obtained using the Bruxoff device were read automatically using the Bruxmeter^®^ (OT Biolettonica in Turin, Italy) software (SW bruxmeter 2.0.2.4).

The calibration of the device was carried out by the manufacturer at the time of its purchase. If the instructions in the user’s manual for the device are followed, the device is designed to last. However, it is recommended by the manufacturers that after 5 years of use, it should be sent to the manufacturer for control.

The electromyographic recordings of the masseter muscle (with myoelectric activity exceeding 0.25 s) were selected for the motor activity score at the oro-facial level [3].

The amplitude limit value was set at 10% of MVC (maximum voluntary contraction) activity when the patient was awake. Thus, motor episodes at the oro-facial level separated by 3 s intervals were recognized as RMMA (rhythmic masticatory muscle activity) if they corresponded to one of the following three models: phasic (three or more EMG bursts lasting 0.25–2 s each), tonic (EMG activity > 2 s), or mixed episodes (both types of activity) [3].

Based on the device, we recorded data regarding (Figure 2):-Number of episodes of bruxism per night (SB/night);-Number of masseter muscle contractions during one night (MC/night);-Number of phasic contractions (phasic RMMA), tonic contractions (tonic RMMA), and mixed contractions (mixed RMMA);-Episodes of bruxism per hour (SB/h);-The patient’s sleep time.

With this device, in addition to data on bruxism, we obtained data on each patient’s heart rate.

The statistical analysis was performed using GraphPad Prism 9 for Mac version 9.3.1. The statistical significance was set at a *p* < 0.05. The parameters taken into account s were the mean (M), median (Me), standard deviation (SD±), and confidence interval (CI 95%).

After entering the values recorded into the database, the null hypotheses were formulated:

**H01.** 
*there are no statistically significant differences regarding the studied parameters between the study group and the control group;*


**H02.** 
*between the studied parameters, within the study group, there is no statistically significant correlation;*


**H03.** 
*there are no statistically significant differences for the studied parameters between the female and the male group;*


Following the Kolmogorov–Smirnov test, it was established that in the studied group, the distribution was non-Gaussian (non-parametric). The tests used for statistical analysis were the following: Mann–Whitney and Spearman.

## 3. Results

The gender distribution of the study group was 50% female (18) and 50% male (18) with a mean age of 10.08 ± 3.451. In the control group, all participants were male with a mean age of 12.47 ± 5.617.

The results of the descriptive statistics for the study group and the control group are presented in Table 1 and Table 2.

The results obtained after applying the Mann–Whitney test to compare the recorded values in the study and control group are represented in Table 3 and Figure 3.

The results demonstrated the statistical differences between all the studied parameters in the control and study groups (Table 3 and Figure 3). Following these results, the null hypothesis H01 was rejected.

Within the study group, to observe whether there was a correlation between the studied parameters, Spearman’s test was applied. No statistically significant correlation was found between the heart rate and MC/n, phasic RMMA, tonic RMMA, and mixed RMMA. The results are represented in Table 4 and Figure 4.

After interpreting the statistical results following the application of Spearman’s correlation test, the null hypothesis formulated H02 was partially rejected.

The study group was divided into two subgroups according to gender, with each subset comprising 18 participants. Descriptive statistics according to the gender of the study participants are presented in Table 5.

The Mann–Whitney test was used to compare the studied parameters according to the gender of the study participants, and the results are presented in Table 6 and Figure 5.

The heart rate, MC/n, and tonic RMMA values showed no statistically significant differences between the female and male study groups (Table 6, Figure 5b,c,e). Thus, the null hypothesis H03 was partially rejected.

## 4. Discussion

In this study, Bruxoff measurements were recorded for a minimum duration of 60 min over three consecutive nights. Over time, a series of portable devices have been developed for the recording and electromyographic monitoring of masseter and temporalis muscle activity during sleep to reduce costs, which would be high in the case of using polysomnography and would also require considerable time consumption. Castroflorio et al. considered rhythmic masticatory muscle activity to be observed in 60% of the general population as physiological masticatory muscle activity during sleep. Thus, portable devices that only measure sEMG activity tend to overestimate the diagnosis of nocturnal bruxism [8,11]. Contrary to this, our results show a significant correlation between the number of masseter muscle contractions and the number of nocturnal bruxism episodes. Not all devices designed to measure oromotor activity during sleep have shown optimal validity if polysomnographic results are considered the gold standard. The absence of audio/video recordings being made at the same time as the polysomnographic recordings may represent a factor that led to a decrease in the validity of the reference standard for the diagnosis of nocturnal bruxism. Currently, the criteria for diagnosing nocturnal bruxism in a laboratory specially created for monitoring during sleep include the presence of specific noises represented by teeth grinding in at least two episodes of bruxism [15].

Studies by Lavigne et al. reported a sudden change in cardiac and respiratory activity and a specific brain activation immediately before a nocturnal bruxism episode [16]. In our study, there was no significant change in heart rate with the appearance of contractions specific to bruxism. Thus, according to Lavigne’s study, formal analysis of the sEMG activity from the masseter muscle and heart rate could represent an excellent solution to improve the reliability of portable devices for the diagnosis of nocturnal bruxism.

A thorough examination of the masticatory muscles and observation of the symptoms of bruxism in the oral cavity is the basis of the diagnosis of bruxism. Each patient qualified for the study group had muscle pain and tenderness symptoms, bruxism-specific lesions in the oral cavity on examination, and confirmed teeth grinding at night. Subsequently, patients who qualified for the control group had no muscle pain on examination, specific signs of oral bruxism, or positive self-report for bruxism. As anticipated, the number of bruxism episodes per hour was significantly higher in the study group. All patients with muscle pain and/or symptoms of bruxism were found to have bruxism. Interestingly, some patients without masticatory muscle pain and/or bruxism symptoms were also found to have bruxism as assessed by the Bruxoff device, as Klara Saczuk et al. [4] observed in their study.

Further studies with a more significant number of subjects and conducted over a more extended period are needed to confirm the results obtained within our study. However, the portable sEMG/ECG device used in this investigation was found to be reliable for measuring what it purported to measure, namely, the motor activity of the masseter muscles during sleep. These findings are significant because of the need for simple and reliable portable devices for diagnosing nocturnal bruxism in clinical and research areas [17].

Regarding the accuracy of the Bruxoff device tested in this study, while the device was found to have a sensitivity of 100%, its specificity of 76% still leaves room for improvement, as there is a chance of giving a healthy subject a false positive diagnosis of bruxism. Bruxoff’s producers rated its sensitivity at 92% and specificity at 85%. Other studies have also shown a higher sensitivity and specificity of the portable Bruxoff device (92.3% and 91.6%, respectively) and a high correlation and concordance between Bruxoff and PSG readings [4,18]. According to the existing polysomnography (PSG) literature, this software can classify an episode of nocturnal bruxism if the sEMG burst is higher than 10% of MVC activity and if a 20% increase immediately follows it in heart rate from baseline [12]. Code concentric ring systems with electrodes have higher spatial selectivity than traditional detection systems and reduce the invasiveness of electrodes because they are invariant in rotation and reduce EMG crosstalk [15,19].

According to the 2018 International Consensus [1], the cut-off points for establishing the presence or absence of bruxism should not be used in healthy individuals. Instead, masticatory muscle activities related to bruxism should be assessed. Even if recent research shows that bruxism could be connected to musculoskeletal pain, it does not support a direct and clear link between them, but rather suggests a complex approach, considering other risk factors [20].

Research has shown that bruxism affects females more frequently than males [21,22]. In our study, in terms of gender, no statistically significant difference was found between the participants. However, regarding the point of view of the mean of the recorded values, it was higher in the group of female participants. These findings are in line with the results obtained by Manfredini et al., Carra et al., and Cavallo et al., which report no gender differences in sleep bruxism [23,24,25,26,27].

Regarding the number and types of muscle contractions (tonic, phasic, and mixed), the results showed significant differences between the study and the control groups. A study conducted by Deregibus et al. showed the absence of a significant correlation between the number of masseter contractions per hour and the number of nocturnal bruxism episodes per hour [28], which is contradicted by our study. According to several studies from the literature, more tonic activity may be associated with daytime bruxism and morning muscle symptoms [5,29,30]. Therefore, the morning muscle pain and fatigue reported by all patients in the study group were caused by sleep bruxism with tonic contractions [31]. Regarding gender differences in the types of masseteric contractions, no statistically significant difference was observed among females compared to males, but the mean among female subjects was higher than among males.

According to our findings, the Bruxoff device proved efficient for patient screening by monitoring both masseter muscle activity and heart rate. The diagnosis of bruxism based only on surface electromyographic analysis tends to overestimate the existence of nocturnal bruxism [29].

Lavigne et al. investigated the variability of facial motor activity in nocturnal bruxism over 2 months and 7 years using an audio-polysomnographic recording [12]. They found that the diagnosis of nocturnal bruxism remained relatively constant over time in moderate to severe bruxism. Individual variability may be necessary for some patients with nocturnal bruxism.

Klara Saczuk et al. argued that the activity of the masseter muscles monitored using electromyographic recordings is the foundation of the most valid theories regarding the etiology of bruxism [4]. Previous studies have confirmed the potential of Bruxoff as a screening device for patients with sleep bruxism [15,28,32]. Because of this theory, the authors suggested that the use of such portable electromyographic recording devices like Bruxoff (Bioelettronica, Italy), BiteStrip (Alldent, Australia), and GrindCare (Sunstar, Switzerland) should be considered by clinicians dealing with the diagnosis and treatment of bruxism [15,17,28,33,34,35].

In a plot study conducted by Needham et al. that used the Grindcare device, 58% of the participants reported a major reduction in the occurence of the side efects represented by headaches and discomfort of the masticatory muscles in the morning. Their data guide practitioners to use such devices due to the devices’ benefits on oro-facial health [36].

The limitations of our study lie in the small number of participants and the recording duration, as well as the fact that no occlusion relationship registrations were taken into consideration. Another important limitation of the study is represented by the fact that there were only male participants in the control group.

## 5. Conclusions

Within the limitations of the present study, we can conclude that the main advantage of using a Bruxoff device is represented by the fact that, at a much lower cost than polysomnography, it helps to diagnose patients with bruxism in our practice activity. Also, according to the present study, a correlation can be observed between the number of masseter contractions and the number of bruxism episodes during the night, with this parafunction being found in both male and female patients. This device registers the gravity of bruxism because it assesses the intensity and duration of this parafunctional activity in each patient. Timely diagnosis of bruxism and informing the parents/patients to start treatment leads to improving the patient’s quality of life. Also, the establishment of an early treatment protects patients against bruxism’s side effects, helping to maintain both oral and mental health.

## Figures and Tables

**Figure 1 diagnostics-13-03233-f001:**
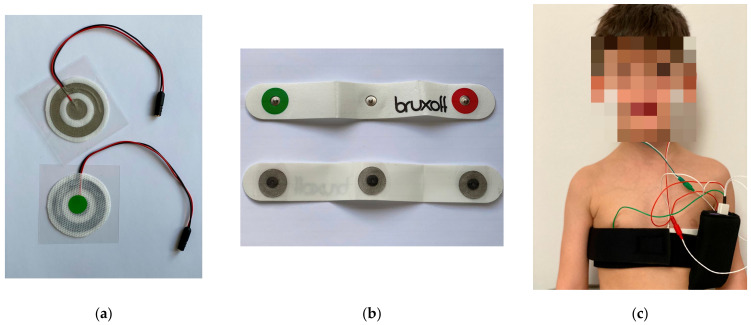
The used electrodes: (**a**) concentric electrodes called Code^®^ (OT Bioelettronica in Turin, Italy); (**b**) concentric electrodes to detect heart rate; (**c**) correct position of the electrodes.

**Figure 2 diagnostics-13-03233-f002:**
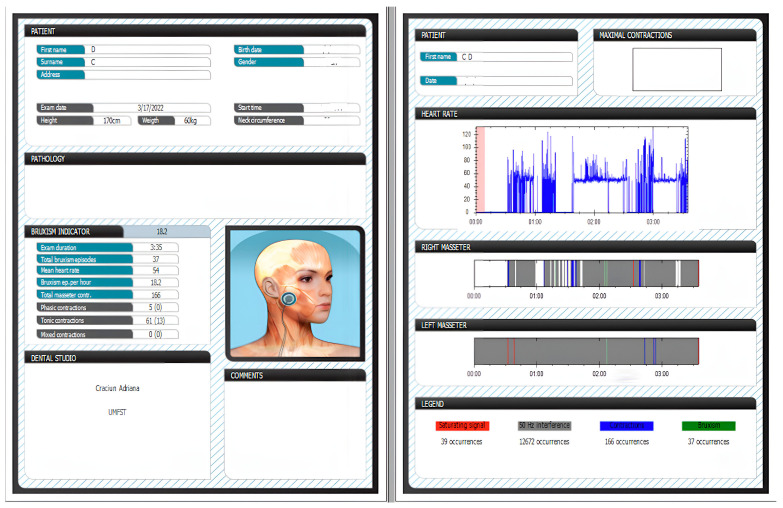
Recorded data using Bruxoff.

**Figure 3 diagnostics-13-03233-f003:**
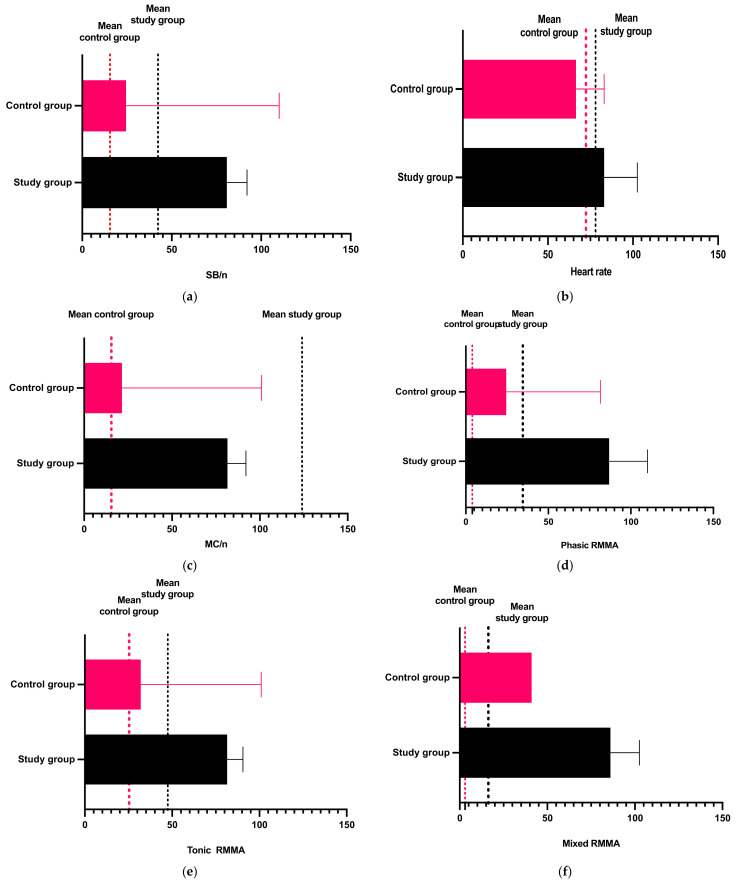
Graphic representation of comparison between control and study group regarding (**a**) bruxism episodes/night; (**b**) heart rate; (**c**) masseter contractions; (**d**) phasic contractions; (**e**) tonic contractions; (**f**) mixed contractions.

**Figure 4 diagnostics-13-03233-f004:**
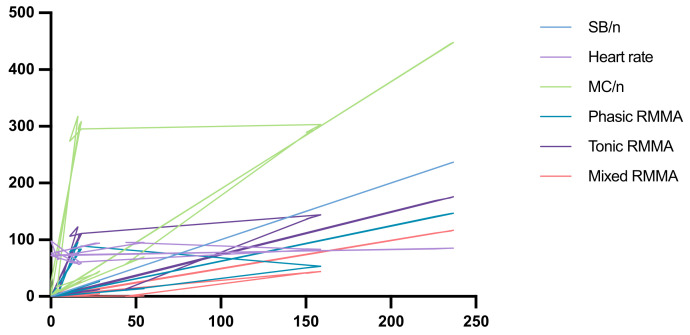
Graphic representation of Spearman’s test.

**Figure 5 diagnostics-13-03233-f005:**
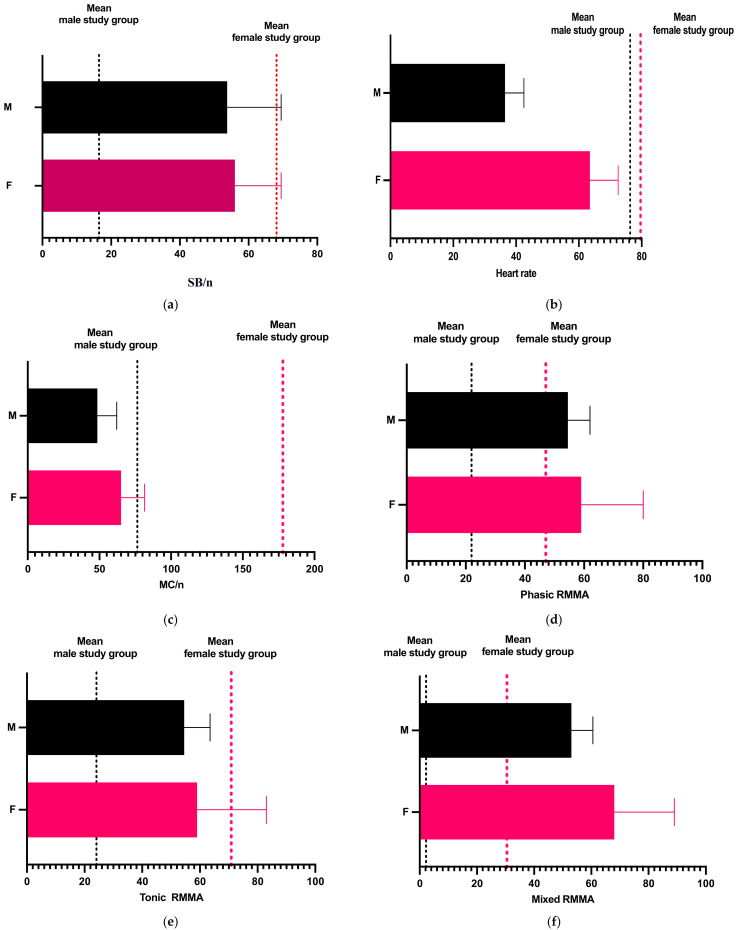
Graphic representation of comparison according to gender regarding (**a**) bruxism episodes/night; (**b**) heart rate; (**c**) masseter contraction; (**d**) phasic contractions; (**e**) tonic contractions; (**f**) mixed contractions.

**Table 1 diagnostics-13-03233-t001:** Descriptive statistics—study group.

	Age	Sleeping Time (min.)	SB/night	Heart Rate	SB/h	MC/night	Phasic RMMA	Tonic RMMA	Mixed RMMA
**Minimum**	5.000	58.00	0.000	57.00	0.000	0.000	0.000	0.000	0.000
**Median**	11.00	71.00	8.000	77.00	4.000	33.00	8.000	8.500	1.000
**Maximum**	16.00	592.0	237.0	98.00	54.00	448.0	147.0	176.0	117.0
**Mean**	10.08	200.6	42.29	77.97	12.30	124.0	34.47	47.47	16.31
**Std. Deviation**	3.451	201.4	71.22	12.97	14.88	154.2	45.73	63.22	32.40
**Lower 95% CI of mean**	8.916	162.2	28.70	75.50	9.462	94.56	25.75	35.41	10.13
**Upper 95% CI of mean**	11.25	239.1	55.87	80.45	15.14	153.4	43.20	59.53	22.49

**Table 2 diagnostics-13-03233-t002:** Descriptive statistics—control group.

	Age	Sleeping Time (min.)	SB/night	Heart Rate	SB/h	MC/night	Phasic RMMA	Tonic RMMA	Mixed RMMA
**Minimum**	5.000	60.00	0.000	54.00	0.000	0.000	0.000	0.000	0.000
**Median**	15.00	75.00	0.000	74.00	0.000	0.000	0.000	0.000	0.000
**Maximum**	19.00	455.0	50.00	86.00	24.60	225.0	15.00	83.00	15.00
**Mean**	12.47	179.4	15.47	72.33	5.467	59.27	3.867	25.33	3.000
**Std. Deviation**	5.617	152.0	19.62	10.38	8.378	79.84	5.375	32.13	5.625
**Lower 95% CI of mean**	9.356	133.7	9.573	69.21	2.950	35.28	2.252	15.68	1.310
**Upper 95% CI of mean**	15.58	225.1	21.36	75.45	7.984	83.25	5.482	34.99	4.690

**Table 3 diagnostics-13-03233-t003:** Results of the comparison between the study group and the control group.

Control Group		SB/night	Heart Rate	MC/night	Phasic RMMA	Tonic RMMA	Mixed RMMA
	Study Group						
***p*** **value**		0.0069	0.0228	0.0001	<0.0001	0.0069	0.0017
**Statistic significance**		**	*	***	****	**	**
**Mann-Whitney U**		1769	1863	1499	1301	1782	1710
**Difference between medians**		−8.000	−3.000	−33.000	−8.000	−8.500	−1.000

* significant, ** very significant, *** extremely significant, **** more than extremely significant.

**Table 4 diagnostics-13-03233-t004:** The results of Spearman’s test.

Parameters		r	*p* Value	Summary
**SB/night**	Heart rate	0.1795	0.0631	Weak positive correlation ns
	MC/night	0.8765	<0.0001	Strong positive correlation ****
	Phasic RMMA	0.8754	<0.0001	Strong positive correlation ****
	Tonic RMMA	0.9201	<0.0001	Strong positive correlation ****
	Mixed RMMA	0.8021	<0.0001	Strong positive correlation ****
**Heart rate**	MC/night	−0.0497	0.6095	Negative correlation/no correlation ns
	Phasic RMMA	−0.1821	0.0592	Negative correlation/no correlation ns
	Tonic RMMA	−0.0287	0.7681	Negative correlation/no correlation ns
	Mixed RMMA	−0.1463	0.1309	Negative correlation/no correlation ns
**MC/night**	Phasic RMMA	0.9400	<0.0001	Strong positive correlation ****
	Tonic RMMA	0.9475	<0.0001	Strong positive correlation ****
	Mixed RMMA	0.8580	<0.0001	Strong positive correlation ****
**Phasic RMMA**	Tonic RMMA	0.9448	<0.0001	Strong positive correlation ****
	Mixed RMMA	0.9081	<0.0001	Strong positive correlation ****
**Tonic RMMA**	Mixed RMMA	0.9156	<0.0001	Strong positive correlation ****

**** extremely significant; ns—not significant; r—Spearman’s rank correlation coefficient.

**Table 5 diagnostics-13-03233-t005:** Descriptive statistics according to the gender.

		Minimum	Median	Maximum	Mean	Std. Deviation	Lower 95% CI of Mean	Upper 95% CI of Mean
**Female study group**	Age	6	12	16	11.33	3.009	10.51	12.15
	Sleeping time (min.)	60	239.5	592	293.8	233.9	229.9	357.6
	SB/night	0	10.5	237	68.11	92.57	42.84	93.38
	Heart rate	57	80.5	98	79.61	12.28	76.26	82.96
	SB/h	0	5.75	25.1	10.08	10.36	7.256	12.91
	MC/night	0	157.5	448	177.9	176.6	129.7	226.1
	Phasic RMMA	0	26.5	147	47.06	54.16	32.27	61.84
	Tonic RMMA	0	55	176	70.83	72.78	50.97	90.7
	Mixed RMMA	0	12.5	117	30.44	41.29	19.17	41.71
**Male study group**	Age	5	8.5	14	8.833	3.369	7.914	9.753
	Sleeping time (min.)	58	66	341	107.5	97.37	80.92	134.1
	SB/night	0	8	55	16.46	17.66	11.64	21.28
	Heart rate	58	72	95	76.33	13.55	72.63	80.03
	SB/h	0	4	54	14.52	18.16	9.561	19.47
	MC/night	0	27	318	70.06	104.4	41.56	98.55
	Phasic RMMA	0	8	96	21.89	31.09	13.4	30.37
	Tonic RMMA	0	8.5	123	24.11	40.75	12.99	35.23
	Mixed RMMA	0	1	8	2.167	2.612	1.454	2.88

**Table 6 diagnostics-13-03233-t006:** Results of the comparison between the male and female study groups.

Male Study Group		SB/night	Heart Rate	MC/night	Phasic RMMA	Tonic RMMA	Mixed RMMA
	Female Study Group						
***p*** **value**		0.5457	0.0433	0.0307	0.4243	0.0973	0.0174
**Statistical significance**		ns	*	*	ns	ns	*
**Mann–Whitney U**		1359	1130	1107	1328	1193	1089
**Difference between medians**		−2.500	−8.500	−130.5	−18.50	−46.50	−11.50

ns—not significant, *—significant.

## Data Availability

The dataset analyzed during the study are available from the first author on request.

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
