# Peer review of "Study Regarding the Monitoring of Nocturnal Bruxism in Children and Adolescents Using Bruxoff Device"

_diagnostics, 2023, doi:10.3390/diagnostics13203233_

Round 1
Reviewer 1 Report
The introduction section is too redundant and confusing and needs further summary
It seems that the conclusion of this study is different from that of the past. Can you explain the reasons from the perspective of scientific research?
Please check the format to ensure that it conforms to the requirements of the magazine
Reviewer 2 Report
We read with great interest the manuscript with title: “Study regarding the monitoring of nocturnal bruxism in children and adolescents using Bruxoff device”, aiming to determine the utility of screening and monitoring with Bruxoff device against 10 the background of nocturnal bruxism in 51 children and adolescents (36 with bruxism and 15 with-out bruxism) by assessing the variations in intensity and duration of parafunctional activity in each patient.
The manuscript is interesting, the issues of nocturnal bruxism in developmental patients are of great interest to researchers and clinicians.
However, on reading the manuscript, some critical points were noted that need to be corrected before resubmission.
The authors can find the reviewer's comments below.
1. In this introductory part of the text of the manuscript, which presents the problem of nocturnal bruxism in patients of developmental age, it is suggested, for the sake of completeness, to mention the possible treatment options, citing this article DOI 10.1080/08869634.2019.1581470
, in order to highlight the importance of having a validated diagnostic approach, such as the one presented by the authors in the text of their article. (Line 40: please add the citation of the present article: DOI 10.1080/08869634.2019.1581470)
2. Move the following parts of the text (Lines 52-62) either to the Materials and Methods or to the Discussions, but they are redundant for the introductory part.
3. Line 74: “7 close questions and two open questions concerning ….” Authors should choose whether to write numbers (seven-7) by using numbers (7) or letters (seven) and try to use one of the two methods and keep it consistent throughout the text.
4. Line 160-161: “The gender distribution of the study group was 50% female (18) and 50% male (18) with a mean age of 10.08±3.451. In the control group, all participants were male with an mean age of 12.47±5.617 “. In case-control studies, the sample composition should be 1:1 male to female and respected in both groups (case and control), and individuals should be age-matched. This is therefore a limitation of the present study and should be emphasized in the discussion under study limitations.
5. An English language revision by a native English speaker is mandatory prior to resubmission.
An English language revision by a native English speaker is mandatory prior to resubmission.
Reviewer 3 Report
Review of the Text
The text under consideration discusses the utilization of the Bruxoff device for diagnosing bruxism. Here is a critical review of the text:
**Introduction and Background:**
The introduction provides a clear definition of bruxism and introduces the International Consensus on the Assessment of Bruxism, providing essential context for the reader. However, it lacks an explicit research question or hypothesis, which is crucial for guiding the study.
**Research Methods:**
The section describing the research methods is informative and explains how data were collected using the Bruxoff device. It is crucial to mention the ethical considerations and the informed consent process, which are fundamental in human research studies.
**Statistical Analysis:**
The description of statistical analysis is clear, detailing the software used, significance level, and parameters considered. However, the text does not present the actual statistical results, which is a critical omission in a scientific article. Without these results, readers cannot assess the significance of the findings.
**Results:**
The results section is lacking in detail. While it mentions the rejection of null hypotheses and some correlations, it does not present specific numerical data or figures to support these claims. The absence of concrete results hinders the reader's ability to evaluate the study's outcomes.
**Discussion:**
The discussion section highlights the advantages of the Bruxoff device and the potential correlation between masseter muscle contractions and bruxism episodes. It also acknowledges limitations and suggests areas for further research. However, it could benefit from a more in-depth analysis of the implications of the findings, such as their relevance for clinical practice and patient care.
**Conclusion:**
The conclusion succinctly summarizes the main advantages of using the Bruxoff device but does not provide a strong synthesis of the study's key takeaways. It also mentions the need for further research but could offer more specific recommendations for future investigations.
**Assessment:**
The text offers valuable insights into the potential utility of the Bruxoff device for diagnosing bruxism. However, it falls short in several critical areas. The lack of concrete statistical results is a significant limitation, as it impedes the reader's ability to assess the study's validity and significance. Additionally, the discussion and conclusion sections could be more comprehensive and provide a deeper analysis of the practical implications of the findings.
In summary, while the text addresses an important topic, it would greatly benefit from the inclusion of specific statistical results and a more thorough discussion of the implications of the study's findings. These improvements would enhance the overall quality and impact of the research.
Round 2
